# Search on the Replay Buffer:
## Bridging Planning and Reinforcement Learning

**Benjamin Eysenbach**$^{\theta\phi}$**, Ruslan Salakhutdinov**$^{\theta}$**, Sergey Levine**$^{\phi\psi}$
$^{\theta}$CMU, $^{\phi}$Google Brain, $^{\psi}$UC Berkeley
beysenba@cs.cmu.edu

## Abstract

The history of learning for control has been an exciting back and forth between two broad classes of algorithms: planning and reinforcement learning. Planning algorithms effectively reason over long horizons, but assume access to a local policy and distance metric over collision-free paths. Reinforcement learning excels at learning policies and the relative values of states, but fails to plan over long horizons. Despite the successes of each method in various domains, tasks that require reasoning over long horizons with limited feedback and high-dimensional observations remain exceedingly challenging for both planning and reinforcement learning algorithms. Frustratingly, these sorts of tasks are potentially the most useful, as they are simple to design (a human only need to provide an example goal state) and avoid reward shaping, which can bias the agent towards find a sub-optimal solution. We introduce a general-purpose control algorithm that combines the strengths of planning and reinforcement learning to effectively solve these tasks. Our aim is to decompose the task of reaching a distant goal state into a sequence of easier tasks, each of which corresponds to reaching a particular subgoal. Planning algorithms can automatically find these waypoints, but only if provided with suitable abstractions of the environment – namely, a graph consisting of nodes and edges. Our main insight is that this graph can be constructed via reinforcement learning, where a goal-conditioned value function provides edge weights, and nodes are taken to be previously seen observations in a replay buffer. Using graph search over our replay buffer, we can automatically generate this sequence of subgoals, even in image-based environments. Our algorithm, search on the replay buffer (SoRB), enables agents to solve sparse reward tasks over one hundred steps, and generalizes substantially better than standard RL algorithms.[1]

## 1  Introduction

How can agents learn to solve complex, temporally extended tasks? Classically, planning algorithms give us one tool for learning such tasks. While planning algorithms work well for tasks where it is easy to determine distances between states and easy to design a local policy to reach nearby states, both of these requirements become roadblocks when applying planning to high-dimensional (e.g., image-based) tasks. Learning algorithms excel at handling high-dimensional observations, but reinforcement learning (RL) – learning for control – fails to reason over long horizons to solve temporally extended tasks. In this paper, we propose a method that combines the strengths of planning and RL, resulting in an algorithm that can plan over long horizons in tasks with high-dimensional observations.

Recent work has introduced goal-conditioned RL algorithms (Pong et al., 2018; Schaul et al., 2015) that acquire a single policy for reaching many goals. In practice, goal-conditioned RL succeeds at

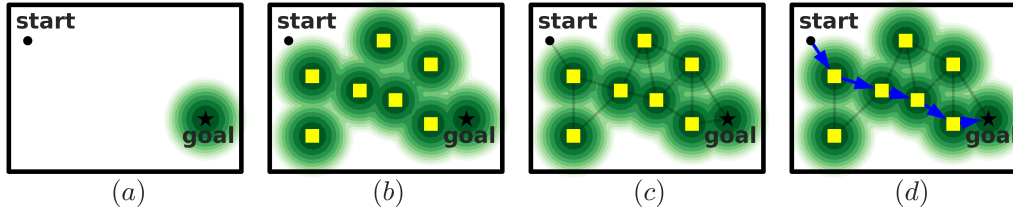

Figure 1: **Search on the Replay Buffer:** (a) Goal-conditioned RL often fails to reach distant goals, but can successfully reach the goal if starting nearby (inside the green region). (b) Our goal is to use observations in our replay buffer (yellow squares) as waypoints leading to the goal. (c) We automatically find these waypoints by using the agent's value function to predict when two states are nearby, and building the corresponding graph. (d) We run graph search to find the sequence of waypoints (blue arrows), and then use our goal-conditioned policy to reach each waypoint.

reaching nearby goals but fails to reach distant goals; performance degrades quickly as the number of steps to the goal increases (Levy et al., 2019; Nachum et al., 2018). Moreover, goal-conditioned RL often requires large amounts of reward shaping (Chiang et al., 2019) or human demonstrations (Lynch et al., 2019; Nair et al., 2018), both of which can limit the asymptotic performance of the policy by discouraging the policy from seeking novel solutions.

We propose to solve long-horizon, sparse reward tasks by decomposing the task into a series of easier goal-reaching tasks. We learn a goal-conditioned policy for solving each of the goal-reaching tasks. Our main idea is to reduce the problem of finding these subgoals to solving a shortest path problem over states that we have previous visited, using a distance metric extracted from our goal-conditioned policy. We call this algorithm Search on Replay Buffer (SoRB), and provide a simple illustration of the algorithm in Figure 1.

Our primary contribution is an algorithm that bridges planning and deep RL for solving long-horizon, sparse reward tasks. We develop a practical instantiation of this algorithm using ensembles of distributional value functions, which allows us to *robustly* learn distances and use them for *risk-aware* planning. Empirically, we find that our method generates effective plans to solve long horizon navigation tasks, even in image-based domains, without a map and without odometry. Comparisons with state-of-the-art RL methods show that SoRB is substantially more successful in reaching distant goals. We also observe that the learned policy generalizes well to navigate in unseen environments. In summary, graph search over previously visited states is a simple tool for boosting the performance of a goal-conditioned RL algorithm.

## 2 Bridging Planning and Reinforcement Learning

Planning algorithms must be able to (1) sample valid states, (2) estimate the distance between reachable pairs of states, and (3) use a local policy to navigate between nearby states. These requirements are difficult to satisfy in complex tasks with high dimensional observations, such as images. For example, consider a robot arm stacking blocks using image observations. Sampling states requires generating photo-realistic images, and estimating distances and choosing actions requires reasoning about dozens of interactions between blocks. Our method will obtain distance estimates and a local policy using a RL algorithm. To sample states, we will simply use a replay buffer of previously visited states as a non-parametric generative model.

### 2.1 Building Block: Goal-Conditioned RL

A key building block of our method is a goal-conditioned policy and its associated value function. We consider a goal-reaching agent interacting with an environment. The agent observes its current state $s \in \mathcal{S}$ and a goal state $s_g \in \mathcal{S}$. The initial state for each episode is sampled $s_1 \sim \rho(s)$, and dynamics are governed by the distribution $p(s_{t+1} \mid s_t, a_t)$. At every step, the agent samples an action $a \sim \pi(a \mid s, s_g)$ and receives a corresponding reward $r(s, a, s_g)$ that indicates whether the agent has reached the goal. The episode terminates as soon as the agent reaches the goal, or after $T$ steps, whichever occurs first. The agent's task is to maximize its cumulative, *undiscounted*, reward. We use an off-policy algorithm to learn such a policy, as well as its associated goal-conditioned Q-function

and value function:

$$Q(s, a, s_g) = \mathbb{E}_{\substack{s_1 \sim \rho(s), a_t \sim \pi(a_t | s_t, s_g) \\ s_{t+1} \sim p(s_{t+1} | s_t, a_t)}} \left[ \sum_{t=1}^{T} r(s_t, a_t, s_g) \right], \qquad V(s, s_g) = \max_a Q(s, a, s_g)$$

We obtain a policy by acting greedily w.r.t. the Q-function: $\pi(a \mid s, s_g) = \arg\max_a Q(s, a, s_g)$. We choose an off-policy RL algorithm with goal relabelling (Andrychowicz et al., 2017; Kaelbling, 1993b) and distributional RL (Bellemare et al., 2017)) not only for improved data efficiency, but also to obtain good distance estimates (See Section 2.2). We will use DQN (Mnih et al., 2013) for discrete action environments and DDPG (Lillicrap et al., 2015) for continuous action environments. Both algorithms operate by minimizing the Bellman error over transitions sampled from a replay buffer $\mathcal{B}$.

## 2.2 Distances from Goal-Conditioned Reinforcement Learning

To ultimately perform planning, we need to compute the *shortest path distance* between pairs of states. Following Kaelbling (1993b), we define a reward function that returns -1 at every step: $r(s, a, s_g) \triangleq -1$. The episode ends when the agent is sufficiently close to the goal, as determined by a state-identity oracle. Using this reward function and termination condition, there is a close connection between the Q values and shortest paths. We define $d_{\mathrm{sp}}(s, s_g)$ to be the shortest path distance from state $s$ to state $s_g$. That is, $d_{\mathrm{sp}}(s, s_g)$ is the expected number of steps to reach $s_g$ from $s$ under the optimal policy. The value of state $s$ with respect to goal $s_g$ is simply the negative shortest path distance: $V(s, s_g) = -d_{\mathrm{sp}}(s, s_g)$. We likewise define $d_{\mathrm{sp}}(s, a, s_g)$ as the shortest path distance, conditioned on initially taking action $a$. Then Q values also equal a negative shortest path distance: $Q(s, a, s_g) = -d_{\mathrm{sp}}(s, a, s_g)$. Thus, goal-conditioned RL on a suitable reward function yields a Q-function that allows us to estimate shortest-path distances.

## 2.3 The Replay Buffer as a Graph

We build a weighted, *directed* graph directly on top of states in our replay buffer, so each node corresponds to an observation (e.g., an image). We add edges between nodes with weight (i.e., length) equal to their predicted distance, using $d_\pi(s_1, s_2)$ as our estimate of the distance using our current Q-function. While, in theory, going directly to the goal is always a shortest path, in practice the goal-conditioned policy will fail to reach distant goals directly (See Fig. 6.). We will therefore ignore edges that are longer than MAXDIST, a hyperparameter:

$$\mathcal{G} \triangleq (\mathcal{V}, \mathcal{E}, \mathcal{W}) \qquad \text{where} \quad \mathcal{V} = \mathcal{B}, \quad \mathcal{E} = \mathcal{B} \times \mathcal{B} = \{e_{s_1 \to s_2} \mid s_1, s_2 \in \mathcal{B}\}$$

$$\mathcal{W}(e_{s_1 \to s_2}) = \begin{cases} d_\pi(s_1, s_2) & \text{if } d_\pi(s_1, s_2) < \text{MAXDIST} \\ \infty & \text{otherwise} \end{cases}$$

Given a start and goal state, we temporarily add each to the graph. We add directed edges from the start state to every other state, and from every other state to the goal state, using the same criteria as above. We use Dijkstra's Algorithm to find the shortest path. See Appendix A for details.

## 2.4 Algorithm Summary

After learning a goal-conditioned Q-function, we perform graph search to find a set of waypoints and use the goal-conditioned policy to reach each. We view the combination of graph search and the underlying goal-conditioned policy as a new SEARCHPOLICY, shown in Algorithm 1. The algorithm starts by using graph search to obtain the shortest path $s_{w_1}, s_{w_2}, \cdots$ from the current state $s$ to the goal state $s_g$, planning over the states in our replay buffer $\mathcal{B}$. We then estimate the distance from the current state to the first waypoint, as well as the distance from the current state to the goal. In most cases, we then condition the policy on the first waypoint, $s_{w_1}$. However, if the goal state is closer

---

**Algorithm 1** Inputs are the current state $s$, the goal state $s_g$, a buffer of observations $\mathcal{B}$, the learned policy $\pi$ and its value function $V$. Returns an action $a$.

---

**function** SEARCHPOLICY($s, s_g, \mathcal{B}, V, \pi$)
$\quad s_{w_1}, \cdots \leftarrow$ SHORTESTPATH($s, s_g, \mathcal{B}, V$)
$\quad d_{s \to w_1} \leftarrow -V(s, s_{w_1})$
$\quad d_{s \to g} \leftarrow -V(s, s_g)$
$\quad$**if** $d_{s \to w_1} < d_{s \to g}$ or $d_{s \to g} >$ MAXDIST
$\quad\quad a \leftarrow \pi(a, \mid s, s_{w_1})$
$\quad$**else**
$\quad\quad a \leftarrow \pi(a, \mid s, s_g)$
$\quad$**return** $a$

---

than the next waypoint and the goal state is not too far away, then we directly condition the policy on the final goal. If the replay buffer is empty or there is not a path in $\mathcal{G}$ to the goal, then Algorithm 1 resorts to standard goal-conditioned RL.

# 3 Better Distance Estimates

The success of our SEARCHPOLICY depends heavily on the accuracy of our distance estimates. This section proposes two techniques to learn better distances with RL.

## 3.1 Better Distances via Distributional Reinforcement Learning

Off-the-shelf Q-learning algorithms such as DQN (Mnih et al., 2013) or DDPG (Lillicrap et al., 2015) will fail to learn accurate distance estimates using the $-1$ reward function. The true value for a state and goal that are unreachable is $-\infty$, which cannot be represented by a standard, feed-forward Q-network. Simply clipping the Q-value estimates to be within some range avoids the problem of ill-defined Q-values, but empirically we found it challenging to train clipped Q-networks. We adopt distributional Q-learning (Bellemare et al., 2017), noting that is has a convenient form when used with the $-1$ reward function. Distributional RL discretizes the possible value estimates into a set of bins $B = (B_1, B_2, \cdots, B_N)$. For learning distances, bins correspond to distances, so $B_i$ indicates the event that the current state and goal are $i$ steps

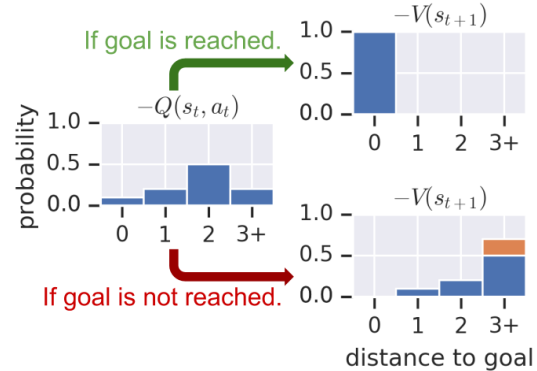

Figure 2: The Bellman update for distributional RL is simple when learning distances, simply corresponding to a left-shift of the Q-values at every step until the agent reaches the goal.

away from one another. Our Q-function predicts a distribution $Q(s_t, s_g, a_t) \in \mathcal{P}^N$ over these bins, where $Q(s_t, s_g, a_t)_i$ is the predicted probability that states $s_t$ and $s_g$ are $i$ steps away from one another. To avoid ill-defined Q-values, the final bin, $B_N$ is a catch-all for predicted distances of at least $N$. Importantly, this gives us a well-defined method to represent large and infinite distances. Under this formulation, the targets $Q^* \in \mathcal{P}^N$ for our Q-values have a simple form:

$$Q^* = \begin{cases} (1, 0, \cdots, 0) & \text{if } s_t = g \\ (0, Q_1, \cdots, Q_{N-2}, Q_{N-1} + Q_N) & \text{if } s_t \neq g \end{cases}$$

As illustrated in Figure 2, if the state and goal are equivalent, then the target places all probability mass in bin 0. Otherwise, the targets are a right-shift of the current predictions. To ensure the target values sum to one, the mass in bin $N$ of the targets is the sum of bins $N-1$ and $N$ from the predicted values. Following Bellemare et al. (2017), we update our Q function by minimizing the KL divergence between our predictions $Q^\theta$ and the target $Q^*$:

$$\min_\theta D_{\text{KL}}(Q^* \parallel Q^\theta) \tag{1}$$

## 3.2 Robust Distances via Ensembles of Value Functions

Since we ultimately want to use estimated distances to perform search, it is crucial that we have accurate distances estimates. It is challenging to robustly estimate the distance between all $|\mathcal{B}|^2$ pairs of states in our buffer $\mathcal{B}$, some of which may not have occurred during training. If we fail and spuriously predict that a pair of distant states are nearby, graph search will exploit this "wormhole" and yield a path which assumes that the agent can "teleport" from one distant state to another. We seek to use a bootstrap (Bickel et al., 1981) as a principled way to estimate uncertainty for our Q-values. Following prior work (Lakshminarayanan et al., 2017; Osband et al., 2016), we implement an approximation to the bootstrap. We train an ensemble of Q-networks, each with independent weights, but trained on the same data using the same loss (Eq. 1). When performing graph search, we aggregate predictions from each Q-network in our ensemble. Empirically, we found that ensembles were crucial for getting graph search to work on image-based tasks, but we observed little difference in whether we took the maximum predicted distance or the average predicted distance.

## 4    Related Work

*Planning Algorithms*: Planning algorithms (Choset et al., 2005; LaValle, 2006) efficiently solve long-horizon tasks, including those that stymie RL algorithms (see, e.g., Kavraki et al. (1996); Lau and Kuffner (2005); Levine et al. (2011)). However, these techniques assume that we can (1) efficiently sample valid states, (2) estimate the distance between two states, and (3) acquire a local policy for reaching nearby states, all of which make it challenging to apply these techniques to high-dimensional tasks (e.g., with image observations). Our method removes these assumptions by (1) sampling states from the replay buffer and (2,3) learning the distance metric and policy with RL. Some prior works have also combined planning algorithms with RL (Chiang et al., 2019; Faust et al., 2018; Savinov et al., 2018a), finding that the combination yields agents adept at reaching distant goals. Perhaps the most similar work is Semi-Parametric Topological Memory (Savinov et al., 2018a), which also uses graph search to find waypoints for a learned policy. We compare to SPTM in Section 5.3.

*Goal-Conditioned RL*: Goal-conditioned policies (Kaelbling, 1993b; Pong et al., 2018; Schaul et al., 2015) take as input the current state and a goal state, and predict a sequence of actions to arrive at the goal. Our algorithm learns a goal-conditioned policy to reach waypoints along the planned path. Recent algorithms (Andrychowicz et al., 2017; Pong et al., 2018) combine off-policy RL algorithms with goal-relabelling to improve the sample complexity and robustness of goal-conditioned policies. Similar algorithms have been proposed for visual navigation (Anderson et al., 2018; Gupta et al., 2017; Mirowski et al., 2016; Zhang et al., 2018; Zhu et al., 2017). A common theme in recent work is learning distance metrics to accelerate RL. While most methods (Florensa et al., 2019; Savinov et al., 2018b; Wu et al., 2018) simply perform RL on top of the learned representation, our method explicitly performs search using the learned metric.

*Hierarchical RL*: Hierarchical RL algorithms automatically learn a set of primitive skills to help an agent learn complex tasks. One class of methods (Bacon et al., 2017; Frans et al., 2017; Kaelbling, 1993a; Kulkarni et al., 2016; Nachum et al., 2018; Parr and Russell, 1998; Precup, 2000; Sutton et al., 1999; Vezhnevets et al., 2017) jointly learn a low-level policy for performing each of the skills together with a high-level policy for sequencing these skills to complete a desired task. Another class of algorithms (Drummond, 2002; Fox et al., 2017; Şimşek et al., 2005) focus solely on automatically discovering these skills or subgoals. SoRB learns primitive skills that correspond to goal-reaching tasks, similar to Nachum et al. (2018). While jointly learning high-level and low-level policies can be unstable (see discussion in Nachum et al. (2018)), we sidestep the problem by using graph search as a fixed, high-level policy.

*Model Based RL*: RL methods are typically divided into model-free (Schulman et al., 2015a,b, 2017; Williams, 1992) and model-based (Lillicrap et al., 2015; Watkins and Dayan, 1992) approaches. Model-based approaches all perform some degree of planning, from predicting the value of some state (Mnih et al., 2013; Silver et al., 2016), obtaining representations by unrolling a learned dynamics model (Racanière et al., 2017), or learning a policy directly on a learned dynamics model (Agrawal et al., 2016;

| model | real states | multi-step | prediction dimension |
|---|---|---|---|
| state-space | ✓ | ✓ | 1000s+ |
| latent-space | ✗ | ✓ | 10s |
| inverse | ✓ | ✗ | 10s |
| SoRB | ✓ | ✓ | 1 |

Figure 3: Four classes of model-based RL methods. Dimensions in the last column correspond to typical robotics tasks with image/lidar observations.

Chua et al., 2018; Finn and Levine, 2017; Kurutach et al., 2018; Nagabandi et al., 2018; Oh et al., 2015; Sutton, 1990). One line of work (Amos et al., 2018; Lee et al., 2018; Srinivas et al., 2018; Tamar et al., 2016) embeds a differentiable planner inside a policy, with the planner learned end-to-end with the rest of the policy. Other work (Lenz et al., 2015; Watter et al., 2015) explicitly learns a representation for use inside a standard planning algorithm. In contrast, SoRB learns to predict the distances between states, which can be viewed as a high-level inverse model. SoRB predicts a scalar (the distance) rather than actions or observations, making the prediction problem substantially easier. By planning over previously visited states, SoRB does not have to cope with infeasible states that can be predicted by forward models in state-space and latent-space.

## 5    Experiments

We compare SoRB to prior methods on two tasks: a simple 2D environment, and then a visual navigation task, where our method will plan over images. Ablation experiments will illustrate that accurate distances estimates are crucial to our algorithm's success.

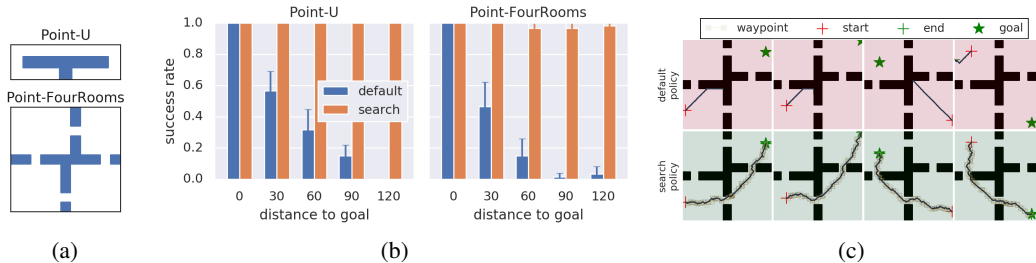

(a)                         (b)                                (c)

Figure 4: **Simple 2D Navigation**: *(Left)* Two simple navigation environments. *(Center)* An agent that combines a goal-conditioned policy with search is substantially more successful at reaching distant goals in these environments than using the goal-conditioned policy alone. *(Right)* A standard goal-conditioned policy (top) fails to reach distant goals. Applying graph search on top of that *same policy* (bottom) yields a sequence of intermediate waypoints (yellow squares) that enable the agent to successfully reach distant goals.

## 5.1 Didactic Example: 2D Navigation

We start by building intuition for our method by applying it to two simple 2D navigation tasks, shown in Figure 4a. The start and goal state are chosen randomly in free space, and reaching the goal often takes over 100 steps, even for the optimal policy. We used goal-conditioned RL to learn a policy for each environment, and then evaluated this policy on randomly sampled (start, goal) pairs of varying difficulty. To implement SoRB, we used exactly the same policy, both to perform graph search and then to reach each of the planned waypoints. In Figure 4b, we observe that the goal-conditioned policy can reach nearby goals, but fails to generalize to distant goals. In contrast, SoRB successfully reaches goals over 100 steps away, with little drop in success rate. Figure 4c compares rollouts from the goal-conditioned policy and our policy. Note that our policy takes actions that temporarily lead away from the goal so the agent can maneuver through a hallway to eventually reach the goal.

## 5.2 Planning over Images for Visual Navigation

We now examine how our method scales to high-dimensional observations in a visual navigation task, illustrated in Figure 5. We use 3D houses from the SUNCG dataset (Song et al., 2017), similar to the task described by Shah et al. (2018). The agent receives either RGB or depth images and takes actions to move North/South/East/West. Following Shah et al. (2018), we stitch four images into a panorama, so the resulting observation has dimension $4 \times 24 \times 32 \times C$, where $C$ is the number of channels (3 for RGB, 1 for Depth). At the start of each episode, we randomly sample an initial state and goal state. We found that sampling nearby goals (within 4 steps) more often (80% of the time) improved the performance of goal-conditioned

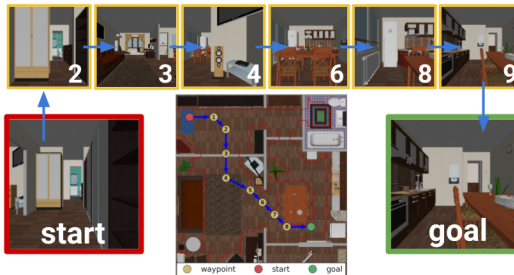

Figure 5: **Visual Navigation**: Given an initial state and goal state, our method automatically finds a sequence of intermediate waypoints. The agent then follows those waypoints to reach the goal.

RL. We use the same goal sampling distribution for all methods. The agent observes both the current image and the goal image, and should take actions that lead to the goal state. The episode terminates once the agent is within 1 meter of the goal. We also terminate if the agent has failed to reach the goal after 20 time steps, but treat the two types of termination differently when computing the TD error (see Pardo et al. (2017)). Note that it is challenging to specify a meaningful distance metric and local policy on pixel inputs, so it is difficult to apply standard planning algorithms to this task.

On this task, we evaluate four state-of-the-art prior methods: hindsight experience replay (HER) (Andrychowicz et al., 2017), distributional RL (C51) (Bellemare et al., 2017), semi-parametric topological memory (SPTM) (Savinov et al., 2018a), and value iteration networks (VIN) (Tamar et al., 2016). SoRB uses C51 as its underlying goal-conditioned policy. For VIN, we tuned the number of iterations as well as the number of hidden units in the recurrent layer. For SPTM, we performed a grid search over the threshold for adding edges, the threshold for choosing the next waypoint along

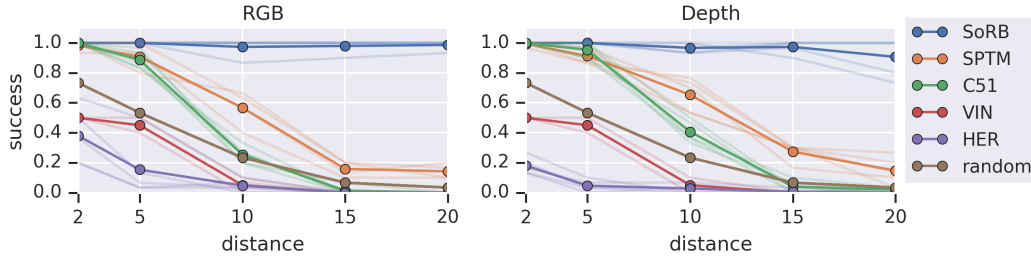

Figure 6: **Visual Navigation**: We compare our method (SoRB) to prior work on the visual navigation environment (Fig. 5), using RGB images *(Left)* and depth images *(Right)* . We find that only our method succeeds in reaching distant goals. *Baselines*: SPTM (Savinov et al., 2018a), C51 (Bellemare et al., 2017), VIN (Tamar et al., 2016), HER (Andrychowicz et al., 2017).

the shortest path, and the parameters for sampling the training data. In total, we performed over 1000 experiments to tune baselines, more than an order of magnitude more than we used for tuning our own method. See Appendix F for details.

We evaluated each method on goals ranging from 2 to 20 steps from the start. For each distance, we randomly sampled 30 (start, goal) pairs, and recorded the average success rate, defined as reaching within 1 meter of the goal within 100 steps. We then repeated each experiment for 5 random seeds. In Figure 6, we plot each random seed as a transparent line; the solid line corresponds to the average across the 5 random seeds. While all prior methods degrade quickly as the distance to the goal increases, our method continues to succeed in reaching goals with probability around 90%. SPTM, the only prior method that also employs search, performs second best, but substantially worse than our method.

### 5.3 Comparison with Semi-Parametric Topological Memory

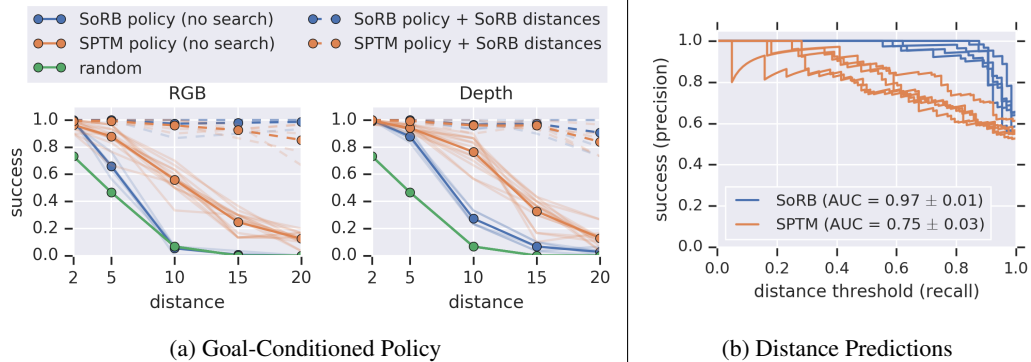

(a) Goal-Conditioned Policy

(b) Distance Predictions

Figure 7: **SoRB vs SPTM**: Our method and Semi-Parametric Topological Memory (Savinov et al., 2018b) differ in the policy used and how distances are estimated. We find *(Left)* that both methods learn comparable policies, but *(Right)* our method learns more accurate distances. See text for details.

To understand why SoRB succeeds at reaching distant goals more frequently than SPTM, we examine the two key differences between the methods: (1) the *goal-conditioned policy* used to reach nearby goals and (2) the *distance metric* used to construct the graph. While SoRB acquires a goal-conditioned policy via goal-conditioned RL, SPTM obtains a policy by learning an inverse model with supervised learning. First, we compared the performance of the RL policy (used in SoRB) with the inverse model policy (used in SPTM). In Figure 7a, the solid colored lines show that, *without search*, the policy used by SPTM is more successful than the RL policy, but performance of both policies degrades as the distance to the goal increases. We also evaluate a variant of our method that uses the policy from SPTM to reach each waypoint, and find (dashed-lines) no difference in performance, likely because the policies are equally good at reaching nearby goals (within MAXDIST steps). We conclude that the difference in goal-conditioned policies cannot explain the difference in success rate.

The other key difference between SoRB and SPTM is their learned distance metrics. When using distances for graph search, it is critical for the predicted distance between two states to reflect whether the policy can successfully navigate between those states: the model should be more successful at

reaching goals which it predicts are nearby. We can naturally measure this alignment using the area under a precision recall curve. Note that while SoRB predicts distances in the range $[0, T]$, SPTM predicts whether two states are reachable, so its predictions will be in the range $[0, 1]$. Nonetheless, precision-recall curves[2] only depend on the ordering of the predictions, not their absolute values. Figure 7b shows that the distances predicted by SoRB more accurately reflect whether the policy will reach the goal, as compared with SPTM. The average AUC across five random seeds is 22% higher for SoRB than SPTM. In retrospect, this finding is not surprising: while SPTM employs a learned, inverse model policy, it learns distances w.r.t. a random policy.

## 5.4 Better Distance Estimates

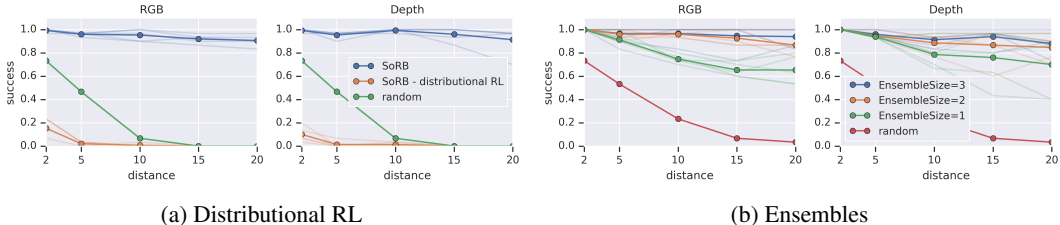

(a) Distributional RL                                        (b) Ensembles

Figure 8: **Better Distance Estimates**: *(Left)* Without distributional RL, our method performs poorly. *(Right)* Ensembles contribute to a moderate increase in success rate, especially for distant goals.

We now examine the ingredients in SoRB that contribute to its accurate distance estimates: distributional RL and ensembles of value functions. In a first experiment, evaluated a variant of SoRB trained without distributional RL. As shown in Figure 8a, this variant performed worse than the random policy, clearly illustrating that distributional RL is a key component of SoRB. The second experiment studied the effect of using ensembles of value functions. Recalling that we introduced ensembles to avoid erroneous distance predictions for distant pairs of states, we expect that ensembles will contribute most towards success at reaching distant goals. Figure 8b confirms this prediction, illustrating that ensembles provide a $10 - 20\%$ increase in success at reaching goals that are at least 10 steps away. We run additional ablation analysis in Appendix C.

## 5.5 Generalizing to New Houses

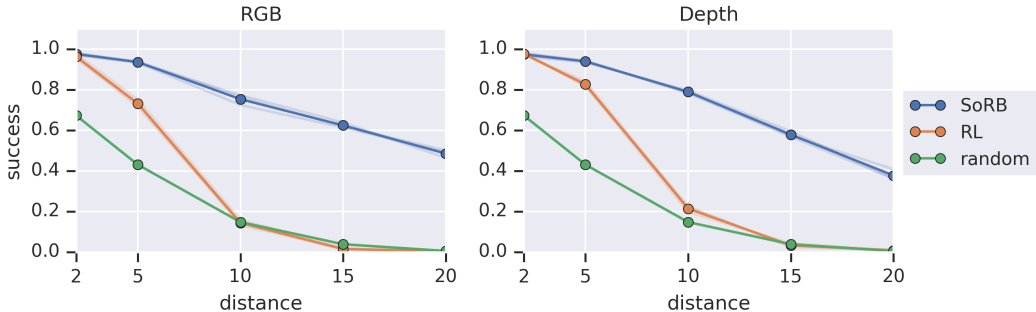

Figure 9: **Does SoRB Generalize?** After training on 100 SUNCG houses, we collect random data in held-out houses to use for search in those new environments. Whether using depth images or RGB images, SoRB generalizes well to new houses, reaching almost 80% of goals 10 steps away, while goal-conditioned RL reaches less than 20% of these goals. Transparent lines correspond to average success rate across 22 held-out houses for each of three random seeds.

We now study whether our method generalizes to new visual navigation environments. We train on 100 SUNCG houses, randomly sampling one per episode. We evaluated on a held-out test set of 22 SUNCG houses. In each house, we collect 1000 random observations and fill our replay buffer with those observations to perform search. We use the same goal-conditioned policy and associated distance function that we learned during training. As before, we measure the fraction of goals reached as we increase the distance to the goal. In Figure 9, we observe that SoRB reaches almost 80% of

goals that are 10 steps away, about four times more than reached by the goal-conditioned RL agent. Our method succeeds in reaching 40% of goals 20 steps away, while goal-conditioned RL has a success rate near 0%. We repeated the experiment for three random seeds, retraining the policy from scratch each time. Note that there is no discernible difference between the three random seeds, plotted as transparent lines, indicating the robustness of our method to random initialization.

## 6    Discussion and Future Work

We presented SoRB, a method that combines planning via graph search and goal-conditioned RL. By exploiting the structure of goal-reaching tasks, we can obtain policies that generalize substantially better than those learned directly from RL. In our experiments, we show that SoRB can solve temporally extended navigation problems, traverse environments with image observations, and generalize to new houses in the SUNCG dataset. Broadly, we expect SoRB to outperform existing RL approaches on long-horizon tasks, especially those with high-dimensional inputs. Our method relies heavily on goal-conditioned RL, and we expect advances in this area to make our method applicable to even more difficult tasks. While we used a stage-wise procedure, first learning the goal-conditioned policy and then applying graph search, in future work we aim to explore how graph search can improve the goal-conditioned policy itself, perhaps via policy distillation or obtaining better Q-value estimates. In addition, while the planning algorithm we use is simple (namely, Dijkstra), we believe that the key idea of using distance estimates obtained from RL algorithms for planning will open doors to incorporating more sophisticated planning techniques into RL.

**Acknowledgements**: We thank Vitchyr Pong, Xingyu Lin, and Shane Gu for helpful discussions on learning goal-conditioned value functions, Aleksandra Faust and Brian Okorn for feedback on connections to planning, and Nikolay Savinov for feedback on the SPTM baseline. RS is supported by NSF grant IIS1763562, ONR grant N000141812861, AFRL CogDeCON, and Apple. Any opinions, findings and conclusions expressed in this material are those of the authors and do not necessarily reflect the views of NSF, AFRL, ONR, or Apple.

## Footnotes

[1]Run our algorithm in your browser: `http://bit.ly/rl_search`

[2]We negate the distance prediction from SoRB before computing the precision recall curve because small distances indicate that the policy should be more successful.

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
