[Supplementary Material · sorb-appendix.pdf]

## A  Efficient Shortest Path Computation

Our policy solves a shortest path problem every time it recomputes a new waypoint. Naïvely running Dijkstra's algorithm to compute a shortest path among the states in our active set $\mathcal{B}$ requires $O(|\mathcal{B}|^2)$ queries of our value function. While the search algorithm itself is fast, it is expensive to evaluate the value function on each pair of states at every time step. In our implementation (Algorithm 2), we amortize this computation across many calls to the policy. We periodically periodically evaluate the value function on each pair of nodes in the replay buffer, and then used the Floyd Warshall algorithm to compute the shortest path between all pairs. This takes $O(|\mathcal{B}|^3)$ time, but only $O(|\mathcal{B}|^2)$ calls to the value function. Let $D \in \mathbb{R}^{|\mathcal{B}| \times |\mathcal{B}|}$ be the resulting matrix storing the shortest path distances between all pairs of states in the active set. Now, given a start state $s$ and goal state $g$, the shortest path distance is

**Algorithm 2** Inputs are the current state $s$, the goal state $g$, the replay buffer $\mathcal{B}$, and the value function $V$. Returns the length and first waypoint of the shortest path.

---

**function** SHORTESTPATH$(s, s_g, \mathcal{B}, V)$
    // Matrices: $D_\pi, D_{\mathcal{B} \to \mathcal{B}}, D_{s \to s_g} \in \mathbb{R}^{|\mathcal{B}| \times |\mathcal{B}|}$
    // Vectors: $D_{s \to \mathcal{B}}, D_{\mathcal{B} \to g} \in \mathbb{R}^{|\mathcal{B}|}$
    $D_\pi \leftarrow -V(\mathcal{B}, \mathcal{B})$          ▷ cached
    $D_{\mathcal{B} \to \mathcal{B}} \leftarrow$ FLOYDWARSHALL$(D_\pi)$   ▷ cached
    $D_{s \to \mathcal{B}} \leftarrow -V(s, \mathcal{B})$
    $D_{\mathcal{B} \to g} \leftarrow -V(\mathcal{B}, g)$
    $D_{s \to g} \leftarrow D_{s \to \mathcal{B}} + D_{\mathcal{B} \to \mathcal{B}} + (D_{\mathcal{B} \to g})^T$
    $s_{w_1} \leftarrow \arg\min_{u,v \in \mathcal{B}} D_{s \to g}$
    **return** $s_{w_1}$

---

$$d_{\text{sp}}(s, g) = \min \left( \min_{u,v \in \mathcal{T}} d(s, u) + D[u, v] + d(v, g), d(s, g) \right)$$

This computation requires $O(|\mathcal{B}|)$ calls to the value function, substantially better than the $O(|\mathcal{B}|^2)$ calls required with the naïve implementation.

## B  Environments

We used two simple navigation environments, Point-U and Point-FourRooms, shown in Figure 4a. In both environments, the observations are the location of the agent, $s = (x, y) \in \mathbb{R}^2$. The agent's actions $a = (dx, dy) \in [-1, 1]^2$ are added to the agents current position at every time step. We tuned the environments so that the goal-conditioned algorithm (which we will use as a baseline) would perform as well as possible. Observing that the agent would get stuck at corners, we modified the environment to automatically add Gaussian noise to the agents action. The resulting dynamics were

$$s_{t+1} = \text{proj}(s_t + a_t + \epsilon_t) \quad \text{where} \quad \epsilon_t \sim \mathcal{N}(0, \sigma^2)$$

where `proj()` handles collisions with walls by projecting the state to the nearest free state. We used $\sigma^2 = 1.0$ for Point-U, and $\sigma^2 = 0.1$ for the (larger) Point-FourRooms environment.

### B.1  Visual Navigation

We ran most experiments on SunCG house `0bda523d58df2ce52d0a1d90ba21f95c`. We repeated all experiments on SunCG house `0601a680273d980b791505cab993096a`, with nearly identical results. We manually choose houses using the following criteria (1) single story, (2) no humans, and (3) included multiple rooms to make planning challenging. During training, we sampled "nearby" goal states (within 4 steps) for 80% of episodes, and sampled goals uniformly at random for the remaining 20% of episodes. We tuned these parameters to make goal-conditioned RL work as well as possible. We implemented goal-relabelling [21, 4], choosing between the (1) originally sampled goal, the (2) current state, and (3) a future state in the same trajectory, each with probability 33%. The agent's actions space was to move North/South/East/West. Observations were panoramic images, created by concatenating the first-person views from each of the cardinal directions. We used ensembles of 3 value functions, each with entirely independent weights. For all neural networks conditioned on both the current observation and the goal observation, we concatenated the current observation and goal observation along their last channel. For RGB images, this resulted in an input with dimensions $H \times W \times 6$. For depth images, the concatenated input had dimension $H \times W \times 2$.

# C    Ablation Experiments

(a) **Replay buffer size**                    (b) **Maximum edge length**

Figure 10: **Sensitivity to Hyperparameters**: *(Left)* While we used a buffer of 1000 observations for most of our experiments, decreasing the buffer size has little effect on the method's success rate. *(Right)* When constructing our graph, we ignore edges that are longer than some distance, MAXDIST. We find that this hyperparameter is important to the success of our method.

Because SoRB plans over a fixed replay buffer, one potential concern is that performance might degrade if the replay buffer is too small. To test this concern, we ran an experiment varying the size of the replay buffer. As shown in Figure 10a, decreasing the replay buffer by a factor of 10x led to no discernible drop on performance. While we do expect performance to drop if we further decrease the size of the replay buffer, the requirement of storing 100 states (even high-resolution images) seems relatively minor. In a second ablation experiment, we varied the MAXDIST hyperparameter that governs when we stop adding new edges to the graph. As shown in Figure 10b, SoRB is sensitive to this hyperparameter, with values too large and too smaller leading to worse performance. When the MAXDIST parameter is too small, graph search fails to find a path to the goal state. As we increase MAXDIST, we increase the probability of underestimating the distance between pairs of states. We expect that improvements in uncertainty quantification in RL will improve the stability of our method w.r.t. this hyperparameter.

# D    Hyperparameters

Unless otherwise noted, all baselines use the same hyperparameters as our method. Unless otherwise noted, parameters were noted tuned.

 **D.1 Search on the Replay Buffer**

| parameter | value | comments |
|---|---|---|
| learning rate | 1e-4 | Initially tried 1e-3 but failed to converge. Lower values also worked, but training took longer. Same for actor and critic. |
| training iterations | 1 million environment steps | Performance changed little after 200k steps |
| batch size | 64 | |
| train steps per environment step | 1:1 | |
| random steps at start of training | 1000 | |
| NN architecture (images) | Conv(16, 8, 4) + Conv(32, 4, 4) + FC(256) | Same for depth and RGB images. |
| optimizer | Adam | We used the default Tensorflow settings for $\beta_1, \beta_2, \epsilon$. Same for actor and critic. |
| MaxDist | 3 | See Figure 10 |
| replay buffer size (training) | 100k | |
| replay buffer size (search) | 1k | See Figure 10 |
| gamma / discount | 1 | |
| $\epsilon$, exploration parameter for discrete actions | 0.1 | Used for visual navigation task |
| OU-stddev, OU-damping, exploration parameter for continuous actions | 1.0, 2.0 | Used for didactic 2D navigation |
| reward scale factor | 0.1 | Tuned for the DDPG baseline on the didactic 2D navigation environment. |
| target network update frequency | every 5 steps | |
| target network update rate ($\tau$) | 0.05 | |

Figure 11: Hyperparameters for SoRB

 **D.2 Semi-Parametric Topological Memory**

We first tuned the $l$ parameter on goal-reaching without search. Setting $l$ to the best found value, we performed a massive (over 1000 experiments) grid search over $M$, $s_{\text{reach}}$, and the threshold for adding edges.

| parameter | value | comments |
|---|---|---|
| threshold for adding edges | 0.9 | Tuned over [0.1, 0.2, 0.5, 0.7, 0.9] |
| $s_{\text{reach}}$, threshold for choosing the next waypoint along the shortest path | 0.5 | Tuned over [0.0, 0.1, 0.2, 0.3, 0.4, 0.5, 0.6, 0.7, 0.8, 0.9, 0.95, 1.0] |
| NN architecture | Conv(16, 8, 4) + Conv(32, 4, 4) + FC(256) | Same architecture (but different weights) for the retrival and locomotor networks. |
| $l$, threshold for sampling nearby states in trajectory | 8 | Tuned over [1, 2, 4, 8] |
| $M$, margin between "close" and "far" states | 1 | Tuned over [1, 2, 4] |

Figure 12: Hyperparameters for SPTM [46]

 **D.3   Value Iteration Networks**

| parameter | value | comments |
|---|---|---|
| number of iterations | 50 | Tuned over [1, 2, 5, 10, 20, 50]. Little effect. |
| hidden units in VI block | 100 | Tuned over [10, 30, 100, 300]. Little effect |

Figure 13: Hyperparameters for VIN [58]

# E   Tricks for Learning Distances with RL

1. *Small learning rates*: Especially for the image-based tasks, we found that RL completely failed with using a critic learning rate larger than 1e-4. Smaller learning rates work too, but take longer to converge.

2. *Distributional RL*: The value function update for distributional RL has a particularly nice form when values correspond to distances. Additionally, distributional RL implicitly clips the values, preventing the critic to predict that unreachable states are infinitely far away.

3. *Termination Condition*: Carefully consider whether you set `done = True` at the end of each episode. In our setting the agent received a reward of -1 at each time step, so the value of each state was negative. An optimal agent therefore attempts to terminate the episode as quickly as possible. We only set `done = True` when the agent reached the goal state, not when the maximum number of time steps was reached or when it reached some other absorbing state.

4. *Ensembles of Value Functions*: Predicted distances from a single value function can be inaccurate for unseen (state, goal) pairs. When performing search using these predicted distances, these inaccurately-short predictions result in "wormholes" through the environment, where the agent mistakenly believes that two distant states are actually nearby. To mitigate this, we trained multiple, independent critics in parallel on the same data, and then aggregated predictions from each before doing search. Surprisingly, we found that taking the average predicted distance over the ensemble worked as well as taking the maximum predicted distance. We tried accelerating training by using shared convolutional layers for all critics in the ensemble, but found that this resulted in highly-correlated distant predictions that exhibited the "wormhole" problem.

# F   Failed Experiments

1. *Goal Relabelling*: As mentioned above, we tried to combine our method with off-policy goal relabelling [4, 43]. Surprisingly, we found that this hurt performance of the non-search policy, and had no effect on the search policy.

2. *Lower-bounds on Q-values*: We attempted to use the search path to obtain a lower bound on the target Q-values during training. In the Bellman update, we replaced the distance predicted by the target Q-values with the minimum of (1) the distance predicted by the target Q-network and (2) the distance of the shortest search path. This can be interpreted as a generalization of the single-step lower bound from Kaelbling [21]. Initial experiments showed this approach slowed down learning, and in some cases prevented the algorithm from converging. We hypothesize that Q-learning is must more sensitive to error in the *relative values* of two actions, rather than the *absolute value* of any particular action. While our lower-bound method likely decreased the absolute error, it did not decrease the relative error (and may have even increased it).

3. *TD3-style Ensemble Aggregation*: In our main experiments, we aggregated distance predictions from the ensemble of distributional critics by first computing the expected distance of each critic, and then averaging the predicted means. This approach ignores the fact that our critics are distributional. Inspired by the stability of TD3, we attempted to apply a similar approach to aggregating predictions from the ensemble of distributional critics. The naïve approach of taking the minimum for each atom does not work because the resulting

distribution will not sum to one. Instead, we first compute the cumulative density function (CDF) of each critic and then take the pointwise maximum over the CDFs. Note that critics correspond to negative distance, so the maximum corresponds to being pessimistic. Finally, we convert the resulting CDF back into a PDF and return the corresponding expected distance. While this method has neat connections to second-order stochastic dominance and risk-averse expected utility maximizers [19], we found that it worked quite poorly in practice.