[Reviews · NeurIPS 2019]

Reviewer 1



After reading the author response, I believe my questions have been fully addressed. ========== The proposed method is novel, and the paper is well written. I have a few clarification questions below. Clarification: • In Algorithm 1, why is it necessary to have a second condition d_{s->g} > MAXDIST? Shouldn’t the first condition d_{s->w_1} < d_{s->g} be sufficient? • It would be more helpful if the authors could provide a complete algorithm table that details the interactions between the goal-conditioned RL and the SearchPolicy in Algorithm 1. Typo: • Between lines 91 and 92, inside the parenthesis of d_{\pi}, it should be s_2 instead of d_2.

Reviewer 2



- The first step in the approach involves the offline learning of a goal-conditioned policy and value functions with distributional RL. The learned policy's inputs are a current state, goal state pair (s, s_g) and must choose an action which will reduce the distance to the goal state. The learned value function now also depends on a pair of states and can therefore be used as a prediction of distance between a (s, s_g) pair. The quality of the distance function is central to the subsequent approach. - Next, the method uses the learned distance-predicting value function to induce a weighted directed graph G over the state space. The vertexes in G are taken from the replay buffer, capped to 1K states (e.g., images) although the appendix suggests the method works with smaller replay buffers as well. I'm not entirely sure how the replay buffer is initialized in an entirely new environment at "test" time and I was unable to find any details in the paper describing how the replay buffer is actually constructed during agent evaluation. Perhaps it's sampling the 1K states from the trajectories generated during the RL step? But the replay buffer may also be empty which suggests that it's being constructed from scratch at test time and updated as the agent explores. In which case, how does the agent decide which states to keep if it gets full? Recency or some other utility function over states? Perhaps Algorithm 1 ought to mention how the buffer is constructed and updated at test time. - Next, the problem of navigating to a distant goal state can be decomposed into a sequence of simpler intermediate navigation tasks to waypoints, which are intermediate states in the graph constructed over the replay buffer. This combination of learned behavior (the goal-conditioned policy and value function predictions of distance) with graph-based planning allows the agent to do well on tasks like visual navigation, which have high-dimensional search spaces, sparse goal-based reward and very long-term temporal dependencies (e.g., moving away from the goal in order to find doors and hallways). - Overall, the primary contribution here seems to be a novel method for learning the length of the shortest path (in [O, T]) between any pair of high-dimensional observations as well as a policy for navigating between them, using distributional RL. This makes the method particularly well suited for the task of visual navigation well. I'm not sure if / how the proposed approach might generalize to other types of benchmark tasks (e.g., Atari, MuJoCo, RTS games like StarCraft, etc.). I think the reader would benefit from a better discussion of the types of settings (besides navigation) where this method is applicable. - Experiments on the effect of the replay buffer size in the appendix were informative and a bit surprising to me that performance didn't deteriorate more although the experiment with MAXDIST confirms that the replay buffer is very important to the approach. The question of what constitutes a "good" replay buffer seems rather important to the overall method. A better understanding of how to optimize the replay buffer in terms of computational tradeoffs, which states to include, etc. would be very interesting to read although it does seem out of the scope of this paper. - The experimental section on SORB itself is nicely detailed and the results on the visual navigation task are strong. The baseline is a relatively recent state-of-the-art method so that seems like a strong baseline. Perhaps the sample complexity and computational complexity of SPTM versus SORB could be discussed in more detail. - Given the clear importance of the learned distance to the overall performance shown in Fig 7, I think the empirical section could be strengthed with evaluations of other methods that learn compact / abstract state representations (e.g., [14]). Compact search spaces would confer computational benefits if nothing else. Overall, studying how compact representations of the state might might compare when used inside graph search seems like a nice way to evaluate just how much utility is added by the distributional RL component of the overall approach. - Overall, the description could be improved with more empirical details about the replay buffer (besides the effect of size) as mentioned above. Besides that, the evaluation of SPTM itself is quite rigorous and the ablation studies reveal the key factors impacting performance. With additional experiments on other distance metric methods and a more detailed description of key components, this could be a nice addition to the literature on combining learning and planning. UPDATE: I thank the authors for their detailed response. After reading the other reviews and the authors response, I reiterate my score.

Reviewer 3



Post rebuttal: My suggestions/comments were not addressed in the rebuttal, so I keep my score as is. --------------------------- This paper focuses on the navigation problems in RL setting and proposes to represent the environment with a graph where the states in the reply buffer become nodes and the low-level RL policy connect the states as the edges. The high-level planner then finds the subgoal by solving the shortest path problem on the graph. Strength: + Building graph with RL value functions is very novel and provides a new way to model the environment and do the planning. + The technique is solid and the paper is well-written. + The paper discussed both useful practice and also contained fail experiments. Weakness: - Although the method discussed by the paper can be applied in general MDP, the paper is limited in navigation problems. Combining RL and planning has already been discussed in PRM-RL~[1]. It would be interesting whether we can apply such algorithms in more general tasks. - The paper has shown that pure RL algorithm (HER) failed to generalize to distance goals but the paper doesn't discuss why it failed and why planning can solve the problem that HER can't solve. Ideally, if the neural networks are large enough and are trained with enough time, Q-Learning should converge to not so bad policy. It will be better if the authors can discuss the advantages of planning over pure Q-learning. - The time complexity will be too high if the reply buffer is too large. [1] PRM-RL: Long-range Robotic Navigation Tasks by Combining Reinforcement Learning and Sampling-based Planning

[Author Response · NeurIPS 2019]

We thank the reviewers for their valuable feedback. Please see the following for the response, and we will make
corresponding modifications in the revised paper.

R2 and R3 had questions about the **time complexity** of our method. The only additional cost of our method over
standard RL is running graph search, which must be done once per evaluation episode. On an 8-core desktop with
32GB of RAM, it takes 65 seconds to estimate the $\mathcal{O}(|V|^2)$ pairwise distances between the 1000 observations in the
replay buffer. As noted in Appendix A, this computation can be amortized across many goal-reaching tasks. Given
that pairwise distances, it takes 0.029 seconds to compute the shortest path (using Dijkstra's Algorithm, which runs in
$\mathcal{O}(|V|^2)$). In terms of **sample complexity**, we found that SoRB required more samples to converge than SPTM, likely
because dynamic programming (the optimization problem for SoRB) is more challenging than supervised learning
(the optimization problem for SPTM). Lastly, we agree with R2 that the construction of "good" replay buffers is an
important problem to explore.

**Reviewer 1:**

• $d_{s \to g} > \textsc{MaxDist}$ – For most distant goals, the shortest path does not go through the waypoint, so the distance
directly to the goal is almost always shorter. However, always conditioning on the goal would be problematic, because
the goal-conditioned policy often fails to reach distant goals directly (See Fig. 6). We will clarify this in Section 2.3.

**Reviewer 2:**

• **Replay buffer**: For the generalization experiments, we initially seeded the replay buffer with 1,000 observations
taken from random states and actions in the new environment. We will clarify this in Alg. 1.

• **Where should SoRB excel?** We expect SoRB to outperform existing RL approaches on long-horizon tasks,
especially those with high-dimensional inputs. The Atari games fit that mould, as does StarCraft II. Many of the
Mujoco tasks in the OpenAI Gym don't require long-horizon reasoning, so we expect little benefit from using SoRB
for environments like Hopper and HalfCheetah. We will add this discussion to Section 6.

• **Comparison with state representations** – We agree that learning compact state representations might boost the
performance of the underlying goal-conditioned agent. While we found Value Iteration Networks (which learn
a representation based on the values of states) performed poorly (see Fig. 6), it is plausible that other methods
(e.g., [14]) would work better. We will include a comparison to [14] for the final version.

**Reviewer 3:**

• **RL and Planning, PRM-RL** – We discussed some prior work that combines RL and planning, including PRM-RL,
on L165 - 167. The key difference between PRM-RL and SoRB is graph construction. Whereas SoRB simply uses
the value function to predict the distance between two states, PRM-RL requires evaluating the policy in the real-world
to determine this distance.[12, Section III.B]

• **Why does Q-learning fail to generalize?** – While, in theory, pure Q-learning should be able to solve any task, in
practice the optimization problem is quite challenging, and only succeeds in learning to reach nearby goals. Our
method generalizes better than Q-learning because, while both are solving a dynamic programming problem at the
high-level, planning provides an exactly solution, while Q-learning gives an approximate solution.

• We will include a comparison to a hierarchical RL method (e.g., [36]) in the final version.

We ran an additional **generalization experiment**. The only difference from Fig. 9 in Section 5.5 is that we now
normalize input images to be in [0, 1] (i.e., divide RGB by 255). We found that normalization substantially improved
the generalization results for RGB images, reaching almost 80% of goals 10 steps away, while goal-conditioned RL
reaches less than 20% of these goals. Transparent lines correspond to average success rate across 22 held-out houses for
each of three random seeds.



[Meta-Review · NeurIPS 2019]

The paper presents a general-purpose control algorithm combining planning and RL to solve tasks with sparse rewards or with long horizon. This algorithm is novel and interesting. The three reviewers agree that the contributions presented here should be published at the conference. The rebuttal helped solving most clarification issues. The reviewers also suggest various ways to further improve the manuscript. These include: - A more detailed discussion on the types of tasks the method could efficiently solve. - A discussion on how the replay buffer could be designed and optimized. - A more precise description of the algorithm and of the experiments.